 eLIFE

# Rheotaxis facilitates upstream navigation of mammalian sperm cells

**Vasily Kantsler[1†‡§], Jörn Dunkel[1†¶], Martyn Blayney[2], Raymond E Goldstein[1*]**

[1]Department of Applied Mathematics and Theoretical Physics, University of Cambridge, Cambridge, United Kingdom; [2]Science, Bourn Hall Clinic, Cambridge, United Kingdom

**Abstract** A major puzzle in biology is how mammalian sperm maintain the correct swimming direction during various phases of the sexual reproduction process. Whilst chemotaxis may dominate near the ovum, it is unclear which cues guide spermatozoa on their long journey towards the egg. Hypothesized mechanisms range from peristaltic pumping to temperature sensing and response to fluid flow variations (rheotaxis), but little is known quantitatively about them. We report the first quantitative study of mammalian sperm rheotaxis, using microfluidic devices to investigate systematically swimming of human and bull sperm over a range of physiologically relevant shear rates and viscosities. Our measurements show that the interplay of fluid shear, steric surface-interactions, and chirality of the flagellar beat leads to stable upstream spiralling motion of sperm cells, thus providing a generic and robust rectification mechanism to support mammalian fertilisation. A minimal mathematical model is presented that accounts quantitatively for the experimental observations.

**\*For correspondence:**
R.E.Goldstein@damtp.cam.ac.uk

[†]These authors contributed equally to this work

**Present address:** [‡]Department of Physics, University of Warwick, Coventry, United Kingdom; [§]Skolkovo Institute of Science and Technology, Skolkovo, Russia; [¶]Department of Mathematics, Massachusetts Institute of Technology, Cambridge, MA, United States

**Competing interests:** The authors declare that no competing interests exist.

**Reviewing editor**: Anthony A Hyman, Max Planck Institute of Molecular Cell Biology and Genetics, Germany

## Introduction

During their journey from ejaculation to fertilisation, human spermatozoa have to find and maintain the right swimming direction over distances that may exceed their head-to-tail length (~100 μm) by a 1000-fold. On their path to the egg cell, mammalian sperm encounter varied physiological environments and are exposed to a variety of chemical gradients and must overcome counterflows. Whilst chemotactic sensing (*Kaupp et al., 2008*) is assumed to provide important guidance in the immediate vicinity of the ovum (*Spehr et al., 2003*), it is not known which biochemical (*Brenker et al., 2012*) or physical mechanisms (*Winet et al., 1984*) keep the sperm cells on track as they pass through the rugged landscapes of cervix, uterus, and oviduct (*Katz et al., 2005*; *Eisenbach and Giojalas, 2006*; *Suarez and Pacey, 2006*). The complexity of the mammalian reproduction process and not least the lack of quantitative data make it very difficult to assess the relative importance of the various proposed long-distance navigation mechanisms (*Eisenbach and Giojalas, 2006*; *Fauci and Dillon, 2006*), ranging from cervix contractions (*Fauci and Dillon, 2006*; *Suarez and Pacey, 2006*) and chemotaxis (*Kaupp et al., 2008*) to thermotaxis (*Bahat et al., 2003*) and rheotaxis (*Miki and Clapham, 2013*). Aiming to understand not only qualitatively but also quantitatively how fluid-mechanical effects may help steer mammalian spermatozoa over large distances, we report here a combined experimental and theoretical study of sperm swimming in microfluidic channels, probing a wide range of physiologically relevant conditions of shear and viscosity. For both human and bull spermatozoa, we find that their physical response to shear flow, combined with an effective shape-regulated surface attraction (*Kantsler et al., 2013*) and head–tail counter-precession, favours an upstream spiralling motion along channel walls. The robustness of this fluid-mechanical rectification mechanism suggests that it is likely to play a key role in the long-distance navigation of mammalian sperm cells. Thus, the detailed analysis reported below not only yields new quantitative insights into the role of biophysical processes during

**eLife digest** A sperm cell must complete a long and taxing journey to stand a chance of fertilising an egg cell. This quest covers a distance that is thousands of times longer than the length of a sperm cell. It also passes through the diverse environments of the cervix, the uterus and, finally, the oviduct, where there might be an egg to fertilise. How the sperm cells manage to stay on course over this distance is a mystery, although it has been suggested that many different factors, including chemical signals and fluid flow, are involved.

The fluids that the sperm cells travel through are not static. Evidence suggests that contractions of the cervix and uterus help to pump sperm cells along the first part of their journey. However, mucus flows out of the oviduct in the opposite direction to way the sperm cells need to go.

Sperm cells mostly move along the walls of the cervix, uterus, and oviduct. This means that sperm cells must contend with two properties of the fluids they travel through—the viscosity (or 'thickness') of the fluid, and the fact that different parts of the fluid will flow at different speeds, depending on how close it is to the wall ('shear flow').

Kantsler et al. have now used a technique called microfluidics—which involves forcing tiny amounts of liquid to flow through very narrow channels—to study how the movement of human and bull sperm cells along a surface is affected by the viscosity and flow rate of the fluid they are swimming through. The sperm cells were found to swim upstream, moving along the walls of the channels in a spiral movement. This is likely to help the sperm cells to find the egg, because spiralling around the oviduct will increase the chances of meeting the egg.

Kantsler et al. also built a mathematical model that describes how the sperm cells move. Although further work is needed to better understand the role played by chemical signals, understanding how fluid flow and viscosity influence sperm cells could lead to more effective artificial insemination techniques.

mammalian reproduction but could also lead to new diagnostic tools and improved artificial insemination techniques (*Merviel et al., 2010*).

Recent experiments on red abalone (*Riffell and Zimmer, 2007*; *Zimmer and Riffell, 2011*), a large marine snail that fertilises externally, showed that weak fluid flows can be beneficial to the reproduction of these organisms, suggesting that shear flows could have acted as a selective pressure in gamete evolution. In higher organisms, which typically fertilise internally, sperm transport is much more complex and the importance of shear flows relative to chemotaxis, peristaltic pumping, or thermotaxis still poses an open problem as it is difficult to perform well-controlled in vivo studies. The complex uterine and oviduct topography (*Suarez and Pacey, 2006*) and large travelling distances render it unlikely that local chemotactic gradients steer sperm cells during the initial and intermediate stages of the sexual reproduction process. Experimental evidence (*Kunz et al., 1996*) suggests that rapid sperm transport right after insemination is supported by peristaltic pumping driven by muscular contractions of the uterus, but it is not known how sperm navigate in the oviduct. Thermotaxis, the directed response of sperm to local temperature differences, was proposed as a possible long-range rectification mechanism of sperm swimming in rabbits (*Bahat et al., 2003*), but recently questioned (*Miki and Clapham, 2013*) as it is likely to be inhibited by convective currents that form in the presence of temperature gradients. On the other hand, it has long been known that, similar to bacteria (*Marcos et al., 2012*) and algae (*Chengala et al., 2013*), mammalian sperm (*Adolphi, 1905*; *Roberts, 1970*) are capable of performing rheotaxis, by aligning against a surrounding flow (*Marcos et al., 2012*; *Miki and Clapham, 2013*), but this effect has yet to be systematically quantified in experiments (*Fauci and Dillon, 2006*; *Suarez and Pacey, 2006*). Specifically, it is not known at present how sperm cells respond to variations in shear rate and viscosity, and how long they need to adapt to temporal changes in the flow direction. Answering these questions is essential for understanding which physical effects may be important at different stages of the mammalian fertilisation process.

To quantify the swimming strategies of sperm cells under well-controlled flow conditions, we performed a series of microfluidic experiments in cylindrical and planar channels (*Figure 1*), varying

systematically shear rates $\dot{\gamma}$ and viscosities $\mu$ through the physiologically and rheotactically relevant regime, up to $\mu$ = 20 mPa·s which is roughly 10× the viscosity of natural seminal fluid (*Owen and Katz, 2005*). These measurements revealed the interesting result that both human and bull spermatozoa do not simply align against the flow, but instead swim upstream on spiral-shaped trajectories along the walls of a cylindrical channel (*Figure 1A*; and *Video 1*). The previously unrecognised transversal

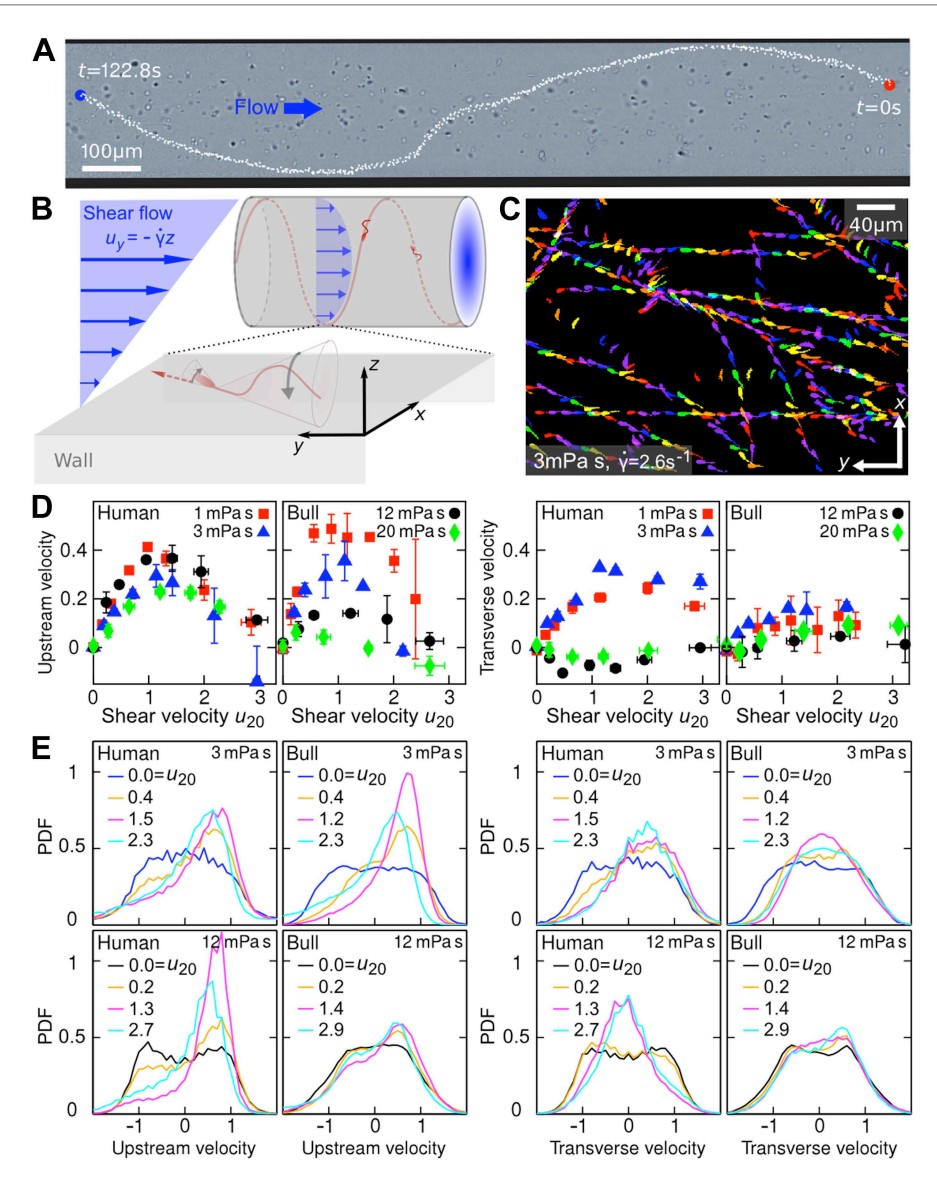

**Figure 1**. Sperm swim on upstream spirals against shear flow. (**A**) Background-subtracted micrograph showing the track of a bull sperm in a cylindrical channel (viscosity $\mu$ = 3 mPa·s shear rate $\dot{\gamma}$ = 2.1s$^{-1}$), channel boundary false-coloured with black, see *Video 1* for raw data. (**B**) Schematic representation not drawn to scale. The conical envelope of the flagellar beat holds the sperm close to the surface (*Kantsler et al., 2013*). The vertical flow gradient exerts a torque that turns the sperm against the flow, but is counteracted by a torque from the chirality of the flagellar wave, resulting in a mean diagonal upstream motion. (**C**) Tracks of bull sperm near a flat channel surface. (**D**) Upstream and transverse mean velocities $\langle v_{y,x} \rangle$ vs shear flow speed $u_{20}$ at 20 μm from the surface for different viscosities. All velocities are normalised by the sample mean speed $v_{0\mu}$ at $\dot{\gamma}$ = 0. For human sperm, in order of increasing viscosity $v_{0\mu}$ = 53.5 ± 3.0, 46.8 ± 3.7, 36.8 ± 3.3, 29.7 ± 3.9 μm/s, and for bull sperm $v_{0\mu}$ = 70.4 ± 11.8, 45.6 ± 4.7, 32.4 ± 4.8, 29.6 ± 4.1 μm/s, where uncertainties are standard deviations of mean values from different experiments. Each data point is an average over >1000 sperms. (**E**) Histograms for selected points in (**D**).

velocity component can be attributed to the chirality of the flagellar beat. The resulting helical swimming patterns enable the spermatozoa to explore collectively the full surface of a cylindrical channel, suggesting that rheotaxis can help sperm to navigate their way through the oviduct and find the egg cell (*Miki and Clapham, 2013*). Using high-speed imaging, we also determined the dynamical response of human and bull spermatozoa to flow reversal at different viscosities, which is essential for understanding how active swimming, rheotaxis, and uterine peristalsis can combine to facilitate optimal sperm transport. To rationalise the experimental observations, we identify below a simple mathematical model that reproduces the main results of our measurements.

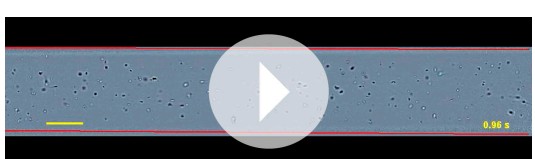

**Video 1**. Human sperm cell swimming on a spiral trajectory (green) against a shear flow in a cylindrical channel (fluid viscosity 3 mPa·s; channel diameter 300 μm; channel boundaries marked in red). Scale bar 100 μm.

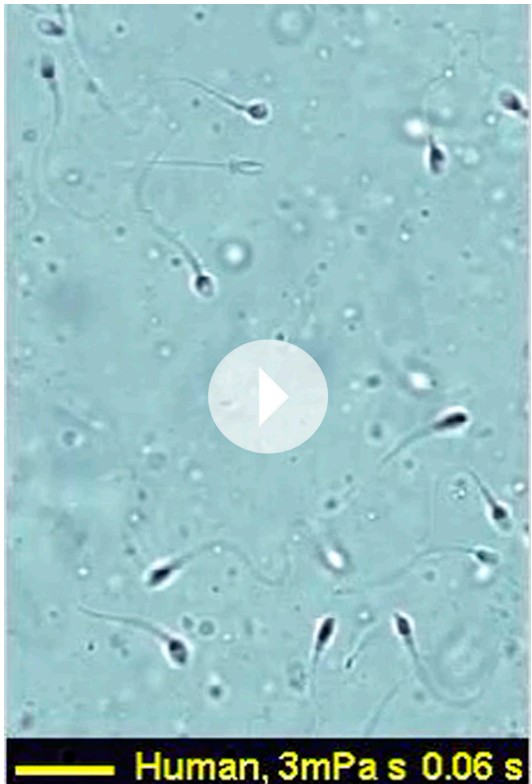

**Video 2**. Human sperm cells swimming in a low-viscosity fluid (3 mPa·s) near the wall of a planar channel. The video shows that, at low viscosity, the flagellar beat of a human sperm cell typically exhibits a considerable chiral component. This follows from the fact that the flagellum never appears as a straight line (in contrast to bull sperms at same viscosity, compare *Video 4*). Scale bar 20 μm.

## Results

### Shear and viscosity dependence

In the experiments, samples of human and bull spermatozoa were injected into microfluidic channels of spherical or rectangular cross-section ('Materials and methods'). The cells were then exposed to well-defined Poiseuille shear flows, corresponding to parabolic flow profiles (*Figure 1B*). Even in the absence of flow, sperm cells tend to accummulate at surfaces (*Rothschild, 1963*; *Denissenko et al., 2012*) due to a combination of steric repulsion (*Kantsler et al., 2013*) and hydrodynamic forces (*Fauci and McDonald, 1995*; *Elgeti et al., 2010*; *Friedrich et al., 2010*; *Gaffney et al., 2011*; *Montenegro-Johnson et al., 2012*). This can be explained by the fact that, in essence, the flagellar beat traces out a cone which, upon collision, aligns with a solid surface, so that the sperm's propulsion vector points into the boundary and the cells become effectively trapped at the surface (*Kantsler et al., 2013*). In the presence of a Poiseuille shear flow, cells close to the channel boundaries experience an approximately linear vertical flow profile, whose slope is given by the shear rate $\dot{\gamma}$ (*Figure 1B*). To quantify the effects of shear rate and viscosity on sperm swimming, we tracked a large number of individual cells (typically $N > 10{,}000$) in planar microfluidic channels (*Figure 1C*) at different shear rates $\dot{\gamma}$, ranging from $0.2\ \text{s}^{-1}$ to $9\ \text{s}^{-1}$, and different dynamic viscosities $\mu$, ranging from 1 mPa·s (that of water) to 20 mPa·s ('Materials and methods'). The cell tracks were then used to reconstruct the velocities of sperm swimming close to the boundary. Mean values and histograms of the upstream and transverse velocity components from those measurements are summarised in *Figure 1D,E*. Since sperm motility depends on viscosity and may vary among different samples, it is advisable to normalise sperm velocities $\boldsymbol{v} = (v_x, v_y)$, that have been measured at different values of $\dot{\gamma}$ and $\mu$, by the mean sample speed $v_{0\mu} = \langle |\boldsymbol{v}| \rangle_\mu$ at zero shear $\dot{\gamma} = 0$, and also to rescale the flow velocity accordingly. *Figure 1D* shows the thus-normalised mean upstream and mean transverse swimming velocities $\langle v_{y,x} \rangle_\mu / v_{0\mu}$ for

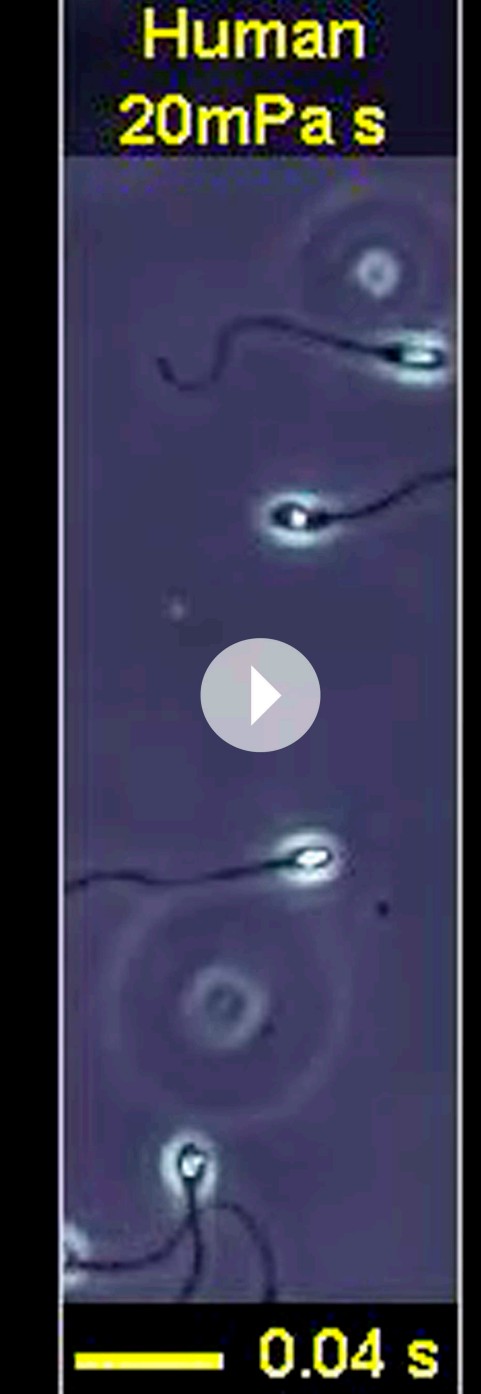

**Video 3**. Human sperm cells swimming in a high-viscosity fluid (20 mPa·s) near the wall of a planar channel. The video shows that, at very high viscosity, the chiral beat component becomes considerably weaker for there now exist instances where the flagellum appears as an almost straight line, indicating that the beat pattern approaches the shape of a planar rotating wave. Scale bar 20 μm.

bull and human spermatozoa as a function of the dimensionless rescaled shear flow speed $u_{20} = \dot{\gamma}\left(2A/v_{0\mu}\right)$, where $A = 10$ μm is the approximate amplitude of a typical flagellar beat (i.e., the maximum distance of the flagellar tip from the surface during a beat is $2A$). The results for the upstream velocity reveal that both human and bull sperm exhibit optimal upstream swimming at rescaled flow speeds $u_{20} \sim 1$, implying that there is an optimal shear regime for the rectification of sperm swimming. Remarkably, however, we also find that, at low viscosities ($\mu \ll 10$ mPa·s), human spermatozoa exhibit a substantial shear-induced transverse velocity component that becomes suppressed at very high viscosities. By contrast, for bull spermatozoa, the mean transverse component is generally weaker and less sensitive to viscosity variations. These statements are also corroborated by the corresponding velocity histograms in *Figure 1E*.

Qualitatively, the above observations for stationary shear flows can be explained as follows. Once a sperm cell has become trapped at a surface, its tail explores, on average, regions of higher flow velocity than the head, resulting in a net torque that turns the head against the flow (*Figure 1B*). This shear-induced rectification is counter-acted by variability in the cells swimming direction. If the shear velocity is too low the orientational 'noise', which is caused by a combination of intrinsic fluctuations in the cells' swimming apparatus, thermal fluctuations, and elastohydrodynamic effects, inhibits upstream swimming, whereas if the shear velocity becomes too large the sperm will simply be advected downstream by the flow, implying that there exists an optimal intermediate shear rate for upstream swimming. Interestingly, we find that the maximum of the upstream velocity decreases more strongly with viscosity for bull sperm than for human sperm (*Figure 1D*). This could be due to differences in cell morphology, as previous numerical studies (*Smith et al., 2011*) for bacterial cells suggest that differences in head shape can substantially alter swimming behavior. Bull sperm have a flatter head than human sperm, which likely suppresses the rotational motion of the cell at high viscosities thus leading to an effectively smaller vertical beat amplitude $A$. This could explain why, at high values of $\mu$, the tail beat of bull sperm becomes essentially two-dimensional and constricted to the vicinity of the surface, so that alignment against the flow becomes less efficient.

To understand the unexpectedly strong transverse velocity component of human sperm at

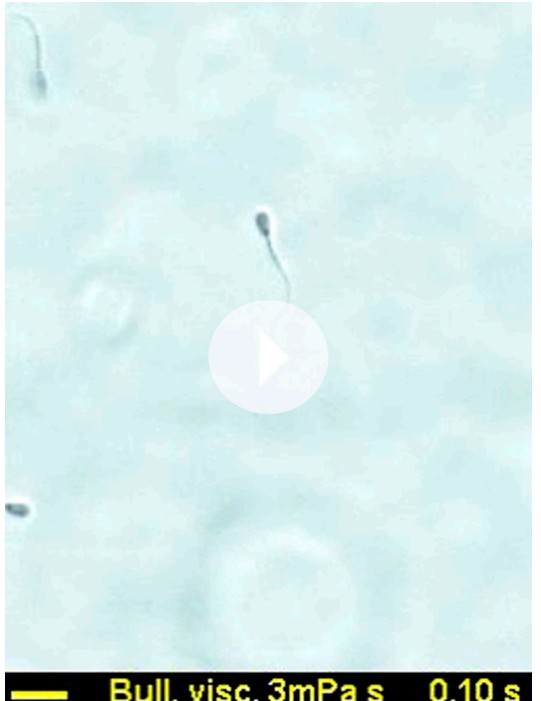

**Bull, visc. 3mPa s    0.10 s**

Video 4. Bull sperm cells swimming in a low-viscosity fluid (3 mPa·s) near the wall of a planar channel. The video shows that, even at low viscosity, the flagellar beat of bull sperm is approximately planar. This follows from the fact that at certain instances the flagellum appears as a line (in contrast to human sperms at same viscosity, compare *Video 2*). Scale bar 20 µm.

$\mu \lesssim 5$ mPa·s, as typical of the seminal fluid (*Owen and Katz, 2005*), it is important to recall that sperm of invertebrae and mammals are known to exhibit different chiral beat patterns depending on environmental conditions (*Gibbons, 1982*; *Ishijima and Hamaguchi, 1993*; *Woolley and Vernon, 2001*; *Smith et al., 2009*), and that shear flows are capable of separating particles along the transverse direction according to their chirality (*Marcos et al., 2009*; *Talkner et al., 2012*). Human sperm exhibit a strong helical beat component at low-to-moderate values of $\mu$ (*Video 2*), but this chirality becomes suppressed at high viscosities (*Video 3*) resulting in more planar wave forms (*Smith et al., 2009*). For comparison, the beat of a bull sperm flagellum is more similar to a rigidly rotating planar wave even at low viscosities (*Video 4*), thus exhibiting a weaker chirality and leading to smaller transverse velocities (*Figure 1D*). Since the flagellar beating pattern can be controlled not only by viscosity but also by changes in calcium concentration (*Ishijima and Hamaguchi, 1993*), higher organisms appear to possess several means for tuning transverse and upstream swimming of sperm.

## Dynamical response

In addition to typically outward directed mucus flow in the oviduct, sperm cells are also exposed to temporally varying flows driven by uterine contractions (*Fauci and Dillon, 2006*; *Suarez and Pacey, 2006*). To probe the dynamical response of sperm to changes in the flow direction, we performed additional experiments, where we tracked the motion of bull and human spermatozoa after a sudden flow reversal at two different viscosities (*Figure 2*; *Videos 5, 6*). In those experiment, sperm were first given time to align against a stationary shear flow, then the flow direction was reversed, $u_y \rightarrow -u_y$, with a switching time <1 s. Upon flow reversal, a sperm cell typically performs a U-turn (*Figure 2A,B*). The characteristic radius of curvature of the trajectory and the typical turning time $\tau$ were found to increase strongly with viscosity. At low viscosity, $\mu \sim 1$ mPa·s, sperm realign rapidly against the new flow direction with a typical response time of $\tau \sim 5$ s to 10 s, and the curvature radius is of the order of one or two sperm lengths $l \sim 60$ µm (*Video 5*). By contrast, at a larger viscosity of $\mu \sim 12$ mPa·s, which is roughly 4× higher than the natural viscosity of the ejaculate, both curvature radius and response time increase by approximately a factor of 5 (*Video 6*). Interestingly, these response times are of the order of typical cervical contractions (*Kunz et al., 1996*), suggesting a possible fine-tuning between muscular activity of the uterus and turning behavior of sperm cells. In particular, immediately after the flow reversal, sperm orientation and flow direction point for a short period of time in approximately the same direction, leading to a momentarily increased transport velocity (see velocity peaks in *Figure 2C*). Thus, by switching flow directions back and forth at an optimal rate, the transport efficiency of an initially rectified sperm population can be enhanced.

## Minimal model

To test whether our understanding of the experimental observations is correct and to provide a basis for future theoretical studies, we used resistive force theory to infer a minimal mathematical model that incorporates the main physical mechanisms discussed above (details are provided in the *Supplementary file 1*). The model assumes that the effective two-dimensional motion of a sperm cell,

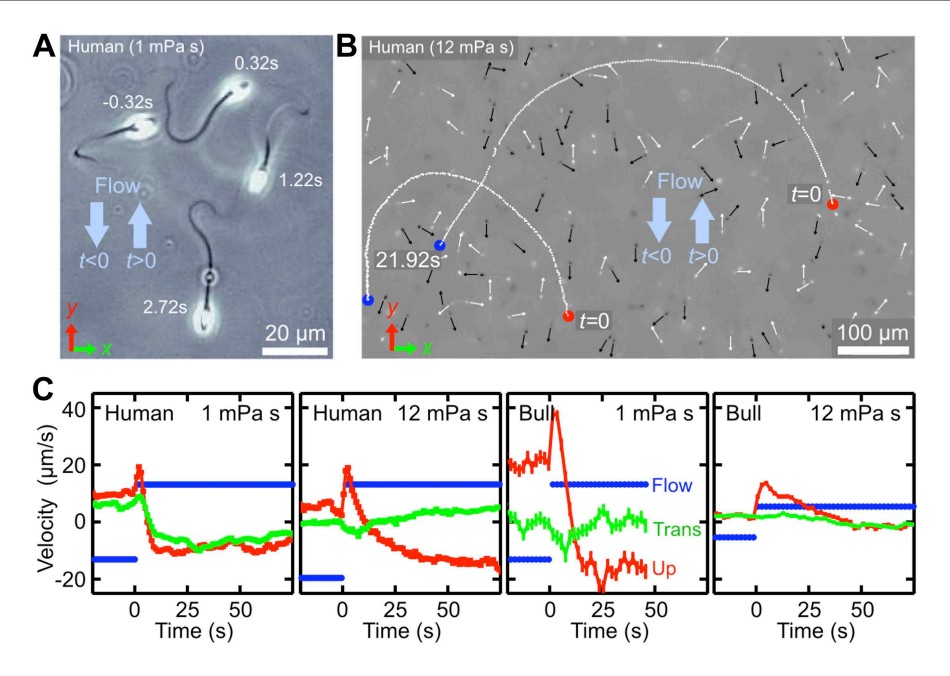

**Figure 2**. Temporal response of sperm cells to a reversal of the flow direction depends sensitively on viscosity. (**A**) At low viscosity, sperm perform sharp U-turns, see also **Video 2**. (**B**) At high viscosity, the typical radius of the U-turns increases substantially (**Video 3**). White/black arrows show orientations of several cells before/after turning. (**C**) Flow velocity at distance 5 μm from the channel surface (blue, 'Flow'), mean upstream velocity $\langle v_y \rangle$ (red, 'Up') and mean transverse velocity $\langle v_x \rangle$ (green, 'Trans') as function of time. The typical response time of sperm cells after flow reversal increases with viscosity. Peaks reflect a short period when mean swimming direction and flow direction are aligned. The time series for human sperm also signal a suppression of the beat chirality at high viscosity, consistent with **Figure 1D**.

that swims close to a surface in the presence of a shear flow can be described in terms of its position vector $\boldsymbol{R}(t) = (X(t), Y(t))$ and its orientation unit vector $\boldsymbol{N}(t) = (N_x(t), N_y(t))$. Focussing on an effective description of the main physical effects and assuming that the flow is in $y$-direction (**Figure 1B**), the equations of motions for $\boldsymbol{R}$ and $\boldsymbol{N}$ read

$$\dot{\boldsymbol{R}} = V\boldsymbol{N} + \sigma \overline{U}\boldsymbol{e}_y, \tag{1}$$

$$\dot{\boldsymbol{N}} = \sigma\dot{\gamma}\alpha \begin{pmatrix} N_x N_y \\ N_y^2 - 1 \end{pmatrix} + \sigma\dot{\gamma}\chi\beta \begin{pmatrix} N_x^2 - 1 \\ N_x N_y \end{pmatrix} + (2D)^{1/2}(\boldsymbol{I} - \boldsymbol{NN}) \cdot \boldsymbol{\xi}(t). \tag{2}$$

**Equation 1** states that the net in-plane velocity $\dot{\boldsymbol{R}}(t)$ of a cell arises from two main contributions: self-swimming at typical speed $V$ in the direction of the cell orientation $\boldsymbol{N}$, and advection by the flow, where $\sigma = \pm 1$ defines the flow direction and $\overline{U} > 0$ the mean flow speed experienced by the cell. As explained in detail in the **Supplementary file 1**, the nonlinear structure of **Equation 2** ensures that the length of the orientation vector $\boldsymbol{N}$ remains conserved, assuming that the change in orientation, $\dot{\boldsymbol{N}}(t)$, is caused by three effects: shear-induced alignment against the flow with rate $\dot{\gamma}\alpha$, where $\alpha > 0$ is numerical factor that encodes geometry of the flagellar beat, shear-and-chirality-induced turning at rate $\dot{\gamma}\beta$ with $\chi \in \{-1, 0, +1\}$ and $\beta > 0$ encoding chirality and shape of the flagellar beat, and variability (**Su et al., 2012**) in the swimming direction, modeled as a Stratonovich-type two-dimensional Gaussian white noise $\xi$ with amplitude $D$ (**Han et al., 2006**). **Equations 1** and **2** were obtained by approximating the flagellum by a rigid conical helix, with the polar geometry of the enveloping cone dictating the mathematical structure of the deterministic turning terms (see **Supplementary file 1** for details of the calculation). The simplifying assumptions underlying **Equations 1** and **2** imply that this minimal model does not accurately capture the dynamics of individual cells at zero shear, as the deterministic terms in

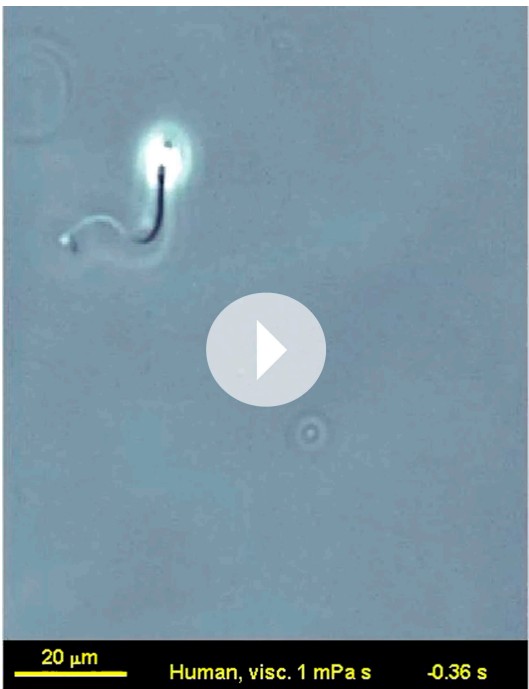

**Video 5**. Reorientation of a human sperm cell swimming in a low-viscosity fluid (1 mPa·s) in a planar channel, after a sudden reversal of the flow direction at time t = 0.

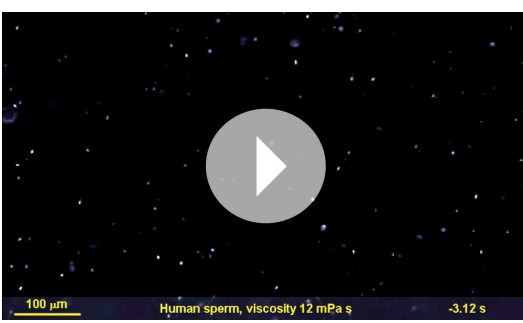

**Video 6**. Reorientation of two human sperm cells, swimming in a high-viscosity fluid in a planar channel, after a sudden reversal of the flow direction at time t = 0.

**Equation 2** neglect the intrinsic curvature of cell trajectories. However, when analyzing the in-plane curvature for a large number of human sperm trajectories (>100,000 sample points from more than 1200 cells) at zero shear, we found a broad distribution of curvatures with a small positive mean curvature of $(5.6 \pm 1.3) \cdot 10^{-4}$ µm at low viscosity (1 mPa·s) and a small negative mean curvature $(-1.9 \pm 0.1) \cdot 10^{-3}$ µm at high viscosity (12 mPa·s), where the different signs are consistent with the observed change in the transverse velocity for human sperm at high viscosity (**Figure 1D**). To account at least partially for these curvature variations, we include in **Equation 2** the Gaussian white noise term. Compared with more accurate models that resolve the details of the flagellar dynamics (**Elgeti et al., 2010**; **Gaffney et al., 2011**), **Equations 1** and **2** provide a strongly reduced description which, however, turns out be sufficient for rationalising our experimental observations (**Figure 3**). Values for $V$ and $\bar{U}$ can be directly estimated from experiments, and sign conventions in **Equation 2** have been chosen such that $\chi = +1$ for human sperm at low viscosity (for weakly chiral bull sperm one can use $\chi = 0$ in a first approximation). The model parameters ($\alpha$, $\beta$, $D$) can be inferred from the experimental data (**Supplementary file 1**). By performing systematic parameter scans, we found that values $\alpha \in [0.2, 0.4]$, $\beta \in [0.05, 0.1]$ and $D \in [0.2, 0.3]$ rad/s yield good quantitative agreement with the experimental results for both stationary flow (**Figure 3A**) and flow reversal (**Figure 3B**), suggesting that the coupling between shear flow and beat chirality dominates the transverse velocity dynamics. We may therefore conclude that, despite some strong simplifications, the effects included in the model capture indeed the main physical mechanisms relevant for understanding sperm motion in shear flow near a surface.

## Discussion

In conclusion, we have reported detailed quantitative measurements of sperm motion in shear flow. Our experimental results show that upstream swimming of mammalian sperm due to rheotaxis is more complex than previously thought. Human sperm cells were found to exhibit a significant transverse velocity component that could be of relevance in the fertilisation process, as the ensuing spiralling motion enables spermatozoa to explore collectively a larger surface area of the oviducts, thereby increasing the probability of locating egg cells. Our theoretical analysis implies that the transverse velocity component arises from a preferred handedness in the flagella beat in the presence of shear flow, in contrast to recent findings for male microgametes of the malaria parasite *Plasmodium berghei* (**Wilson et al., 2013**). Due to the large sample size, our data provide substantial statistical evidence for the hypothesis that mammalian sperm have evolved to achieve optimal upstream swimming near surfaces, possibly exploiting the enhanced fluid production in the female reproductive system during the fertile phase (**Eschenbach et al., 2000**) and after intercourse (**Miki and Clapham, 2013**). The improved

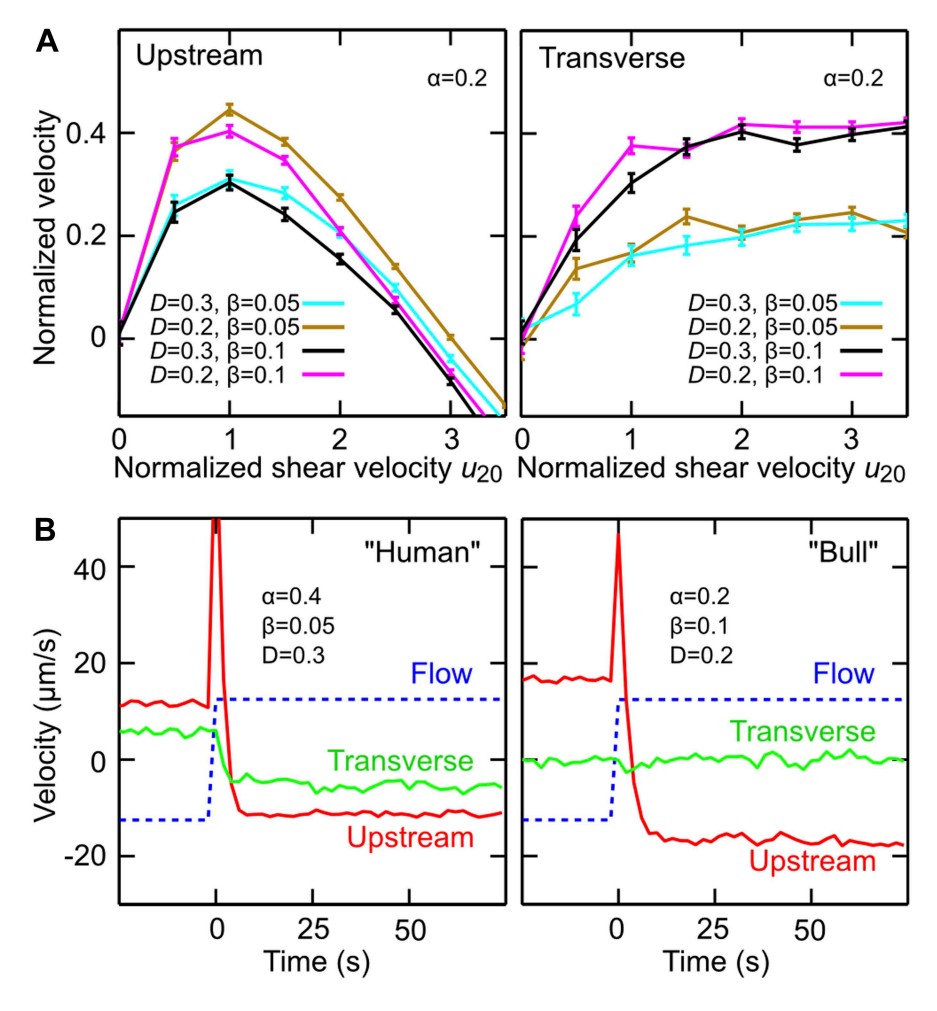

**Figure 3**. Model simulations reproduce main experimental observations. (**A**) Upstream and transverse velocity for different values of the variability (effective noise) parameter $D$ in units rad/s and dimensionless shape factors ($\alpha$, $\beta$). (**B**) Time response of a chiral swimmer with $\chi = +1$ ('Human') and a non-chiral swimmer with $\chi = 0$ ('Bull') to a reversal of the flow direction at time $t = 0$. Blue dashed line shows fluid flow $u_y$ at 5 $\mu$m from the boundary. Simulation parameters ($N = 1000$ trajectories, $A = 10$ $\mu$m, $l = 60$ $\mu$m, $V = 50$ $\mu$m/s) were chosen to match approximately those for viscosity 1 mPa·s in **Figure 2C**.

quantitative knowledge derived from this data may help to design more efficient artificial insemination strategies, for example, by optimising the viscosity and chemical composition of fertilisation media and adjusting injection techniques to maximise upstream swimming of sperm cells. Combined with recent measurements (**Kantsler et al., 2013**), which clarified the importance of flagella-mediated contact interactions for the accumulation of sperm cells at surfaces, the results presented here yield a cohesive picture of the mechanistic and fluid-mechanical (**Friedrich et al., 2010**) aspects of long-distance sperm navigation. Future work should focus on merging these insights with quantitative studies of chemotaxis (**Spehr et al., 2003**; **Kaupp et al., 2008**; **Zimmer and Riffell, 2011**; **Brenker et al., 2012**) to obtain a differentiated understanding of the interplay between physical and chemical factors during various stages of the mammalian reproduction process.

## Materials and methods

### Sperm sample preparation

Cryogenically frozen bull spermatozoa were purchased from Genus Breeding. For each experiment, a bull sperm sample of 250 µl was thawed in a water bath at 37°C for 15 s. Human samples from healthy

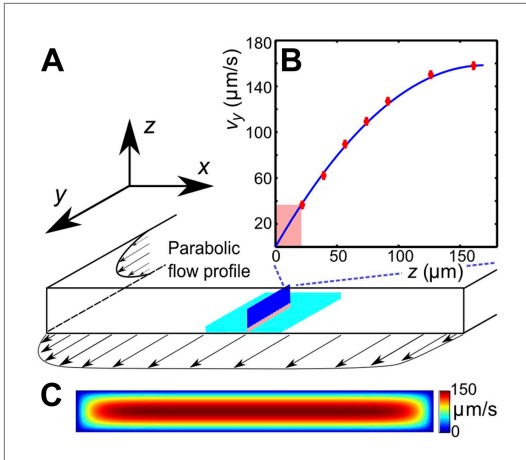

**Figure 4**. (**A**) Schematic of the microfluidic channel and field of view (turquoise region) in the sperm motility measurements. (**B**) Velocity profile at the center of the channel. Red symbols are values of the vertical velocity profile $v_y(z)$ measured by PTV for the flow rate 0.1 µl/s. The solid line shows the theoretically calculated flow profile for the same flow rate. In motility experiments, values for the velocity gradient near the boundary (pink region) were obtained by measuring the flow velocity at 20 µm from the boundary. (**C**) Theoretical 2D flow speed profile in (x,z)-plane at flow rate 0.1 µl/s.

undisclosed normozoospermic donors were obtained from Bourn Hall Clinic. Donors provided informed consent in accordance with the regulations of The University of Cambridge Human Biology Research Ethics Committee. For each experiment, bull and human samples were washed three times by centrifugation at 500 rcf for 5 min with the appropriate medium. The medium for bull spermatozoa contained 72 mM KCl, 160 mM sucrose, 2 mM Na-pyruvate, and 2 mM Na-phosphate buffer at pH 7.4 (*Woolley and Vernon, 2001*). Human sperm medium was based on a standard Earle's Balanced Salt Solution, containing 66.4 mM NaCl, 5.4 mM KCl, 1.6 mM $CaCl_2$, 0.8 mM $MgSO_4$, $N_2H_2PO_4$ 1 mM, $NaHCO_3$ 26 mM, D-Glucose 5.5 mM supplemented with 2.5 mM Na pyruvate and 19 mM Na-lactate pH adjusted to 7.2 by bubbling the medium with $CO_2$. Viscosity of the medium was modified by adding methylcellulose (M0512; Sigma-Aldrich; St. Louis, MO; approximate molecular weight 88,000) at concentrations 0%, 0.2%, 0.4%, 0.5% wt/vol. The absence of circular trajectories at zero-shear implies that the sperm are capacitated (*Miki and Clapham, 2013*).

## Microfluidics

Microfluidic channels were manufactured using standard soft-lithography techniques. The master mould was produced from SU8 2075 (MicroChem Corp.; Newton, MA) spun to a 340 microns thickness layer and exposed to UV light through a high resolution mask to obtain the desired structures. The microfluidic chip containing the channels cast from PDMS (Sylgard 184; Dow Corning; Midland, MI) and bonded to covered glass. The channel has rectangular cross-section of 0.34 × 3 mm. We treated PDMS surfaces of the channels prior the experiment with 10% (wt/vol) Polyethylene glycol (m.w. 8000; Sigma) solution in water for 30 min to avoid adhesion of sperm cells to the walls. Sperm suspension was introduced through inlets with a micro-syringe pump (Harvard Apparatus; Kent, UK) at controlled flow rates of 0.1–40 µl/min. The concentration of the sperm cells in the experiments was kept below 1% volume fraction.

## Microscopy

To identify the swimming characteristics of individual sperm cells, the trajectories were reconstructed by applying a custom-made particle-tracking-velocimetry (PTV) algorithm to image data taken with a Zeiss Axio Observer inverted microscope (20x or 10x objective, 25 fps). The flagella dynamics was captured with a Fastcam SA-3 Photron camera (San Diego, CA; 125 fps, 40x/NA 0.6 objective). Calibration of the velocity profile in the channel was performed by measuring trajectories of fluorescent beads for different distances from the coverslip via PTV. The measured velocity profile is found identical to the calculated values from solving the Stokes equations for the given geometry. Values of the shear rate $\dot{\gamma}$ in the different experiments were reconstructed from the flow velocity at distance 20 µm from the wall (see below).

## Additional experimental information

Effects of viscosity variation and shear-rate variation were studied in experiments that were performed in a rectangular channel with a cross-section 0.34 × 3 mm, by observing sperm motion at lower and upper channel walls. The field of view (normally 800 × 800 µm) was chosen at the middle of the channel (in x-direction), where the in-plane velocity gradient is negligible due to the high aspect ratio of the channel (*Figure 4B*). The $v_y$-velocity profiles, measured along the z-coordinate, were found to be in perfect agreement with the theoretically expected parabolic flow profile for this geometry (*Figure 4B*). The shear rate $\dot{\gamma}$ at a given flow rate was determined from the flow velocity at distance 20 µm from the wall. The depth of field of the objective was <5 µm to ensure that we only observed cells that swam

close to the surface. Trajectories of individual sperm cells were analysed in MATLAB. The sample size in a single experiment exceeds 100,000 velocity vectors, each measurement for a given viscosity and a shear rate was repeated a few times with different sperm samples. Supplemental data tables that summarise the statistical information for each experiment are given in *Supplementary file 2*.

## Additional information

### Funding

| Funder | Grant reference number | Author |
| --- | --- | --- |
| European Research Council | 247333 | Vasily Kantsler, Jörn Dunkel, Raymond E Goldstein |

The funders had no role in study design, data collection and interpretation, or the decision to submit the work for publication.

### Author contributions
VK, Conception and design, Acquisition of data, Analysis and interpretation of data, Drafting or revising the article; JD, REG, Conception and design, Analysis and interpretation of data, Drafting or revising the article; MB, Conception and design, Acquisition of data, Drafting or revising the article

### Ethics
Human subjects: Human samples from healthy undisclosed normozoospermicdonors were obtained from Bourn Hall Clinic. Donors provided informed consent in accordance with the regulations of The University of Cambridge Human Biology Research Ethics Committee, which granted approval to this research under application number HBREC.2013.15.

## Additional files

### Supplementary files
• Supplementary file 1. This file contains a detailed mathematical derivation of the minimal model in *Equations 1* and *2* of the main text, a description of the parameter estimation procedure and a brief summary of numerical methods.

• Supplementary file 2. This file contains data tables that summarise the statistical information for each experiment.

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
