## [Decision Letter]

Thank you for sending your work entitled “Rheotaxis facilitates upstream navigation of mammalian sperm cells” for consideration at *eLife*. Your article has been favorably evaluated by a Senior editor and 3 reviewers, one of whom, Tony Hyman, is a member of our Board of Reviewing Editors.

The Reviewing editor and the other reviewers discussed their comments before we reached this decision, and the Reviewing editor has assembled the following comments to help you prepare a revised submission.

It has been known for a long time that spermatozoa are not just dragged along by a fluid flow, but they tend to swim upstream of the flow, under certain circumstances. A recent paper by Miki et al. in Curr. Biol. (2013) showed that there are fluid flows in the oviduct of mammals for some time after coitus that guide sperm swimming upstream towards the egg. Your paper adds to our knowledge by providing the first quantitative analysis of how mammal spermatozoa swimming is guided by surrounding fluid flows and provides a substantial contribution to the understanding of how sperm are transported to the egg before fertilization. The theory is elegant and the arguments presented are quite clear.

Our main criticism is that in many places you make strong statements without sufficient justification. We would like you to go through your manuscript and consider making adjustments at these places.

For instance in the fifth paragraph of the Introduction section, you state “Human sperm exhibit a strongly helical beat component at low-to-moderate values of μ, but this chirality becomes suppressed at high viscosities due to increased friction (Figure 1). For comparison, the beat of a bull sperm flagellum is more similar to a rigidly rotating planar wave, thus exhibiting a weaker chirality and leading to smaller transverse velocities at physiologically relevant values of μ.” This is a crucial point of the paper and needs to be validated by citations.

There are several statements in the Discussion section that interpret the observations, but it is unclear how strong the arguments presented are. You need to either tone down your statements, or provide better explanations. For example the statement “This is a consequence of the fact that bull sperm have a flatter head which suppresses the rotational motion of the tail ...” is presented as a clear fact. However, I do not see direct evidence for the claim that the head shape is the precise cause for the observation. There are several such problems in the Discussion section that leave the reader with strong statements that remain unclear as to the precise supporting evidence. A second example is “... but the chirality becomes suppressed at high viscosities due to increased friction (Figure 1)”. Figure 1 does not however provide any information about the beat shape, only about the swimming path. Also the “due to increased friction” referring to a changed chirality of beat shape seems to be just a hypothesis.

You do not comment that their swimming paths are straight at zero shear rates. Chiral asymmetry of the beat would allow for a bend in the swimming path that has been observed experimentally (probably for non-capacitated sperm). Why is this possibility not appearing in the simple model? Is there not a measurable radius of curvature of the trajectories observed in the experiments?

There remains some confusion by the symmetries of the terms in [Disp-formula equ2] that are derived in the theory section. Both alignment terms in [Disp-formula equ2] have the same structure and thus should have similar symmetries (one aligns in y-direction, one in x-direction). However, the term proportional to beta was introduced as a chiral term, while the term proportional to alpha is a shear-aligning term that exits for nonchiral elongated objects. This is somewhat puzzling. Also, the shear alignment term usually has a nematic symmetry as shear only provides an axis but no direction. However, the terms in [Disp-formula equ2] describe alignment with a stable direction (not axis), which is a vectorial symmetry. The theory would be more compelling if the underlying symmetries would become clear and their relation to chiral, vector or tensor asymmetries would be transparent.

---

## [Author Response]

The main criticism expressed in the decision letter was that we made several strong statements without sufficient justification. To address these concerns, we have carefully reformulated the corresponding text parts, added several references that provide support for our conclusions and also provide some additional data and videos.

*1) For instance in the fifth paragraph of the Introduction section, you state “Human sperm exhibit a strongly helical beat component at low-to-moderate values of μ, but this chirality becomes suppressed at high viscosities due to increased friction (*Figure 1*). For comparison, the beat of a bull sperm flagellum is more similar to a rigidly rotating planar wave, thus exhibiting a “weaker chirality and leading to smaller transverse velocities at physiologically relevant values of μ.” This is a crucial point of the paper and needs to be validated by citations*.

We now include three additional Videos (now listed as Videos 2, 3 and 4) from our experiments that illustrate representative beat patterns of human and bull sperm cells at different viscosities. Videos 2 and 3 show human sperm at low (3 mPa·s) and high (20 mPa·s) viscosity. At low viscosity (Video 2), the projection of the rotating flagellum never appears as a straight line, implying that the beat pattern has a strong chiral component. By contrast, at high viscosity (Video 3), one can observe instances where the projected flagellum appears as an almost straight line, suggesting that the beat pattern approaches the shape of a rotating planar wave. For comparison, the new Video 4 shows bull sperm at low (3 mPa·s) viscosity. Here the projected beat patterns are qualitatively similar to those of human sperm at high viscosity. We reformulated the corresponding manuscript parts accordingly and also added a reference to an earlier recent experimental study that reports qualitatively similar findings for human sperm cells [DJ Smith et al, Cell Motil. Cytoskeleton 66, 220 (2009)].

*2) … the discussion section is quite unclear. Here are several statements that interpret the observations but it is not clear how strong the arguments presented are. You need to either tone down your statements, or do a better job of explaining them. For example the statement “This is a consequence of the fact that bull sperm have a flatter head which suppresses the rotational motion of the tail ...” is presented as a clear fact. However, I do not see direct evidence for the claim that the head*
*shape is the precise cause for the observation.”*

We have now reformulated this text part, and we also added a reference that emphasizes the potential importance of cell-morphology (in particular, head-shape) for swimming behavior near surfaces [DJ Smith et al, Biophys. J. 100, 2318 (2011)].

*3) There are several such problems in the Discussion section that leave the reader with strong statements that remain unclear as to the precise supporting evidence. A second example is “... but the chirality becomes suppressed at high viscosities due to increased friction (*Figure 1*)”.*
Figure 1
*does not however provide any information about the beat shape, only about the swimming path and the “due to increased friction” referring to a changed chirality of beat shape seems just a hypothesis*.

As stated in our response to point 1), the new Videos 2, 3 and 4 provide some additional visual evidence for the suppression of chirality in the beat patterns of human sperms at higher viscosity. During the revision process we have reformulated this text part and removed the phrase “due to increased friction”*.*

*4) You do not comment that their swimming paths are straight at zero shear rate. Chiral asymmetry of the beat would allow for a bend in the swimming path that has been observed experimentally (probably for non-capacitated sperm). Why is this possibility not appearing in the simple model? Is there not a measurable radius of curvature of the trajectories*
*observed in the experiments?*

We have reanalyzed cell trajectories (>100,000 data points from >1200 cells) and found a broad distribution of curvatures at zero shear for human sperm with a small positive mean curvature of 5.6e-04 +/-1.3e-04 per micron at low viscosity (1mPa·s) and a small negative mean curvature - 1.9e-03 +/- 1e-04 per micron at high viscosity (12mPa·s). The difference in the sign of the mean value is consistent with the observed change of the transverse velocity in Figure 1. This is evident from the derivation in the Supplementary file 1, our simple model cannot account for this effect because it is based on the analytically tractable approximation of a rigid conical helix. To account at least partially for this variability in trajectory curvature, we included in [Disp-formula equ2] the Gaussian white noise term that neglects the small bias due to the shifted mean of the curvature distribution. The reasonable agreement between the simulation results and the experimental data suggests that, even with this simplification, [Disp-formula equ1] and [Disp-formula equ2] can provide a useful minimal description of the upstream/transverse swimming behaviour under shear. We have modified the discussion in the model section of the main text to state the above more clearly.

*5) There remains some confusion by the symmetries of the terms in Equation. (2) that are derived in the theory section. Both alignment terms in*
[Disp-formula equ2]
*have the same structure and thus should have similar symmetries (one aligns in y-direction, one in x-direction). However, the term proportional to beta was introduced as a chiral term, while the term proportional to alpha is a shear-aligning term that exits for nonchiral elongated objects.” This is somewhat puzzling. Also, the shear alignment term usually has a nematic symmetry as shear only provides an axis but no direction. However, the terms in*
[Disp-formula equ2]
*describe alignment with a stable direction (not axis), which is a vectorial symmetry. The theory would be more compelling if the underlying symmetries would become clear and their relation to chiral, vector or tensor asymmetries would be transparent*.

Perhaps the most important difference compared with earlier studies is that we model the flagellum by a conical helix (and not by a cylindrical helix). There is a dominant geometric polarity that determines the leading-order contributions in our model. This is more clearly stated in the substantially extended discussion of the model in the main text.